# Type-II Dirac semimetal stabilized by electron-phonon coupling

Mirko M. Möller[1,2], George A. Sawatzky[1,2], Marcel Franz[1,2] & Mona Berciu[1,2]

There is major interest, in condensed matter physics, in understanding the role of topology: remarkable progress has been made in classifying topological properties of non-interacting electrons, and on understanding the interplay between topology and electron–electron interactions. We extend such studies to interactions with the lattice, and predict non-trivial topological effects in infinitely long-lived polaron bands. Specifically, for a two-dimensional many-band model with realistic electron–phonon coupling, we verify that sharp level crossings are possible for polaron eigenstates, and prove that they are responsible for a novel type of sharp transition in the ground state of the polaron that can occur at a fixed momentum. Furthermore, they result in the appearance of Dirac cones stabilized by electron–phonon coupling. Thus, electron–phonon coupling opens an avenue to create and control Dirac and Weyl semimetals.

[1] Department of Physics and Astronomy, University of British Columbia, Vancouver, BC V6T 1Z1, Canada. [2] Stewart Blusson Quantum Matter Institute, University of British Columbia, Vancouver, BC V6T 1Z4, Canada. Correspondence and requests for materials should be addressed to M.M.Möl. (email: moellerm@phas.ubc.ca) or to M.B. (email: berciu@phas.ubc.ca)

The study of the role played by topology in systems of non-interacting electrons, and in systems with electron–electron interactions, is by now a well-established and robust area of research[1–4]. One recent development is the prediction[5] and experimental observation[6,7] of Type II Dirac/Weyl semimetals. They differ from the ordinary type I Dirac/Weyl cones found, e.g., in graphene, in that they are strongly tilted[5] and have no analog in high-energy physics, because the corresponding particle would break Lorentz invariance. A consequence of the strong tilting is that electron and hole pockets coexist at the Fermi energy and touch at the Dirac/Weyl point, which leads to many interesting properties, e.g., a strong signature in the quantum oscillations of the density of states[8].

The study of the interplay between topological properties and electron-lattice interactions has only very recently started to receive attention. So far, a few issues have been addressed; one is the effect of the topological order of the electronic sector on the properties of phonons, and whether it is possible to experimentally identify a topological state by measuring phonon linewidths or changes in the infrared reflectivity, Raman scattering, etc.[9–12]. Another issue is whether coupling to thermal phonons present at high temperatures can enhance the stability of an electronic topological phase; here there is some disagreement between results obtained from model Hamiltonians[13–18] and from first-principle calculations[19,20]. Density functional theory studies have also asked whether lattice distorsions due to applied pressure or to excitation of specific phonon modes (e.g., by ultrafast laser spectroscopy) can favor a topological phase due to the corresponding change in the lattice symmetry group[21,22]. Throughout, the electron–phonon coupling is assumed to be weak and its vertex depends only the phonon momentum $\mathbf{q}$, but not on the quasiparticle momentum $\mathbf{k}$. Physically, this means that the coupling to the lattice modifies the on-site energy of electrons, but not their hopping integrals.

In weakly doped systems, sufficiently strong electron–phonon coupling significantly renormalizes the properties of the quasiparticle and leads to the formation of a polaron, i.e., the bare particle dressed by a phonon cloud. The single-particle spectrum consists of one or more coherent polaron bands at low energies (the lowest eigenstates), and a broad continuum extending over a wide range of higher energies, in the whole Brillouin zone (BZ). The latter contains finite lifetime eigenstates, due to scattering of the quasiparticle on phonons not bound to the cloud. In previous work addressing electron–phonon coupling, the topologically interesting features (and the Fermi energy) fall within this continuum. What are the consequences of the finite lifetimes on topological properties is still an entirely open question. One available result, in the context of graphene, is that the Dirac point survives such smearing even for very strong electron–phonon coupling, because the vanishing density of states in the bare particle spectrum inhibits scattering, and thus lifetimes close to the Dirac point remain long[23].

When the Fermi energy lies within the infinitely long-lived polaron bands one may be tempted to think of the polarons as being just like bare particles (with renormalized dispersion) and therefore inheriting their properties, but the presence of the phonon clouds does have non-trivial consequences. In models like Holstein[24] and Fröhlich[25,26], where the electron–phonon coupling comes from the modulation of the on-site energy of the particle, additional quasiparticle bands are pushed below the continuum at strong-enough coupling. Apart from the ground state polaron, the most studied excited state is the so-called second bound state in the Holstein model[27,28]. It is now well understood that the fast crossover into the strongly-coupled, small polaron regime, is due to level repulsion between this state and the low-energy polaron[29–32]. As shown in ref. 33, only smooth crossovers are allowed in such models, suggesting that quasiparticle states never cross each other because the phonon clouds always mediate some degree of level repulsion, so the existence of Dirac cones is impossible. Very recently, true transitions have been found in models where the electron–phonon coupling comes from the modulation of the particle hopping[34–36], however what happens here is that the shape of the polaron band changes with increased coupling, so that the miminum moves from the free-particle ground state (GS) momentum to some other value. This may happen continuously[34,35] or discontinuously[36], but again it is not due to level crossing. It is thus not a priori clear if sharp level crossings can occur in polaron spectra.

Here we demonstrate that for an electron–phonon interaction which depends on both $\mathbf{k}$ and $\mathbf{q}$, sharp level crossings do appear in polaron spectra; that they are responsible for a new type of sharp transition in the ground state of the polaron that occurs at a fixed momentum; and that they lead to the appearance of type-II Dirac cones (and presumably of Weyl cones in similar 3D models), whose location in the BZ is controlled by the strength of the electron–phonon coupling. Our work suggests a new pathway for their realization. Of course, direct experimental tuning of the carrier-phonon coupling is difficult, however one can use pump-probe experiments to resonantly excite phonons, which then couple to the carriers[37]. By optically pumping a suitable material, it may therefore be possible to create Dirac cones and to shift their location in the BZ, thus directly controling the topological properties of the material.

## Results

**Stabilization of type II Dirac point.** We now present and discuss the spectrum of the polaron that forms when a hole is doped into a two-dimensional (2D) Lieb lattice, depicted in Fig. 1b. The Hamiltonian, discussed in detail in Methods, describes electrons hopping among the sites of the Lieb lattice coupled to an Einstein phonon mode. As detailed in Methods, there are two relevant electron–phonon couplings: $\alpha$ characterizes the phonon

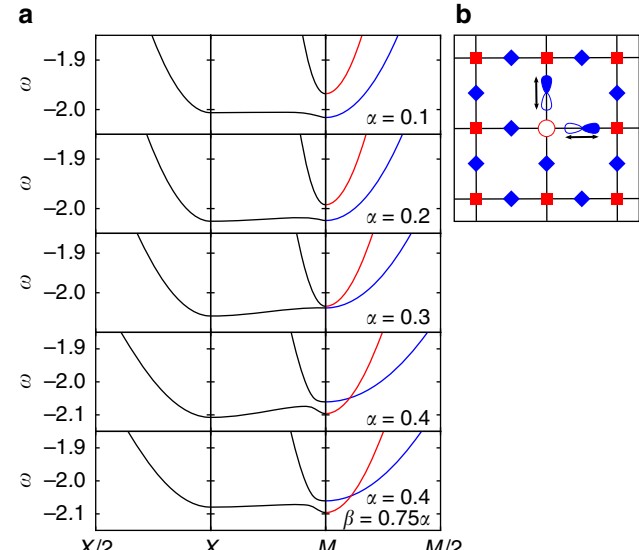

**Fig. 1** Polaron spectrum at various electron–phonon couplings (color online). **a** The infinitely long-lived polaron bands obtained from the spectral function $A^{\gamma,\gamma}(\mathbf{k}, \omega)$. For $M \to M/2$ the symmetry of the bands is indicated in blue for $p_+$ and red for $p_-$. Parameters are $t_{sp} = -1$, $t_{pp} = -0.49$, $\epsilon_s = \epsilon_p = 0$, $\Omega = 1$ and $\alpha = 0.1...0.4$. In the bottom panel $\beta = 0.75\alpha$, in all other panels $\beta = \alpha/2$. **b** The Lieb lattice. Arrows indicate the direction of oscillation of the phonons

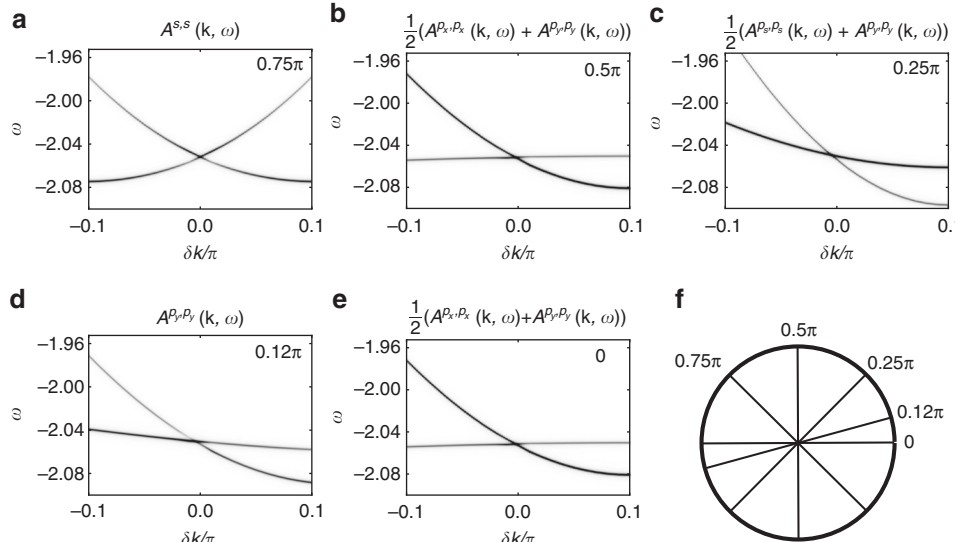

**Fig. 2** Spectral weight cuts through the Dirac point. **a–e** Spectral weight $A^{\gamma,\gamma}(\mathbf{k}, \omega)$ for different cuts through the Dirac point. The projections $\gamma$ were chosen so that both polaron bands are visible. **f** The orientation of the different cuts, parametrized as $(k_x, k_y) = \mathbf{k}_C + \delta k(\cos\phi, \sin\phi)$. Parameters are $t_{sp} = -1$, $t_{pp} = -0.49$, $\epsilon_s = \epsilon_p = 0$, $\Omega = 1$, $\alpha = 0.4$ and $\beta = 0.2$

modulation of the $t_{sp}$ hopping between a central atom and its adjacent ligand atoms, while $\beta$ characterizes the phonon modulation of the $t_{pp}$ hopping between two neighbor ligand atoms.

Our key results are illustrated in Fig. 1, where we plot the evolution of the low-energy part of the polaron spectrum with increasing electron–phonon coupling $\alpha$ for $\beta = \alpha/2$, except in the bottom panel where $\beta = 3\alpha/4$. For an Einstein phonon with energy $\Omega = 1$, the continuum starts at much higher energies than those displayed, thus all shown states are infinitely long-lived polarons. The parameters are chosen such that in the absence of electron–phonon coupling, the lowest bands do not cross (see Methods Section). In the right panels, blue/red colors indicate even/odd character ($p_\pm$) under the $(x, y) \rightarrow (y, x)$ reflection $\hat{\sigma}_{xy}$.

The odd $p_-$ band couples more efficiently to phonons[36] and moves faster toward lower energies with increasing electron–phonon coupling, so that for $\alpha \approx 0.3$ the two bands touch at the $M$ point. For even larger $\alpha$, the $p_-$ band becomes the low-energy band near $M$, and a tilted Dirac point appears where the two bands cross. Its position moves from $M$ toward $\Gamma$ with increasing coupling, e.g., for $\alpha = 0.4$ it is located at $(0.9, 0.9)\pi$.

The $\hat{\sigma}_{xy}$ symmetry only prevents band mixing on the $M - \Gamma$ line; everywhere else we expect to see mixing between the two bands. Indeed, an avoided level crossing is observed at $\approx (1, 0.9)\pi$ on the $M - X$ cut. In Fig. 2 we confirm that the two polaron bands do indeed only cross at a single point in the BZ. Using a k-space grid with a mesh-size of $0.01\pi$ we first find the wavevector $\mathbf{k}_C$ of the Dirac point. The panels show the spectral weight along cuts $(k_x, k_y) = \mathbf{k}_C + \delta k(\cos\phi, \sin\phi)$ which pass through the Dirac point, for various values of $\phi$ illustrated in the bottom right panel. They prove that the crossing does indeed take place at a single point, and also show that the resulting Dirac cone is strongly tilted.

**Sharp polaron transition at fixed momentum.** The final major result is the demonstration of a new type of sharp GS polaron transition. Consider first the top four panels of Fig. 1, where we see that the GS moves from $M$ to $X$ for some $\alpha \leq 0.2$ (for all values of $\alpha$, we scanned the whole BZ; the GS is always either at $M$ or $X$). This looks, therefore, like the discontinuous sharp GS transition found for the 1D version of this model[36], coming from a change in the shape of the lowest polaron band. However, the behavior

here is more interesting and rich than in the 1D model. A hint is provided by the bottom panel, for $\alpha = 0.4$ and $\beta = 0.75\alpha$, that shows the GS back at the $M$ point, but now with odd symmetry.

The full evolution of the GS is illustrated in Fig. 3, where we plot the polaron energies at the $M$ point for both the even and odd bands, plus the lowest value at the $X$ point, vs. $\alpha$. In panels (a) and (b) corresponding to $\beta/\alpha = 0.5$ and 0.75, respectively, we see two discontinuous transitions as the GS jumps from $M$ (even) $\rightarrow X \rightarrow M$ (odd) with increasing $\alpha$. This is already very interesting: such polaron behavior, i.e., a GS momentum that does not evolve monotonically with increased electron–phonon coupling, has not been seen before.

The truly new feature, however, is seen in Fig. 3c: here the GS is always at the $M$ point, but it has a transition from even to odd symmetry as $\alpha$ increases. This is clearly a sharp polaron transition occuring at fixed GS momentum, driven by band crossing. It is thus qualitatively different from the only two other known kinds of sharp polaron transitions[34–36], which are due to a band deformation and are accompanied by a change of the GS momentum. It is interesting to point out that until rather recently, the consensus in the community was that sharp polaron transitions are not possible. This was proved rigorously for **q**-dependent models of the Holstein and Fröhlich type[33]. Our results show that a lot more work is needed to understand all the possible scenarios which can occur for different types of **k** and **q**-dependent electron–phonon couplings.

## Discussion

These results clearly prove that a sharp crossing of two polaron bands is indeed possible in a model with suitable symmetries. While this may seem to be rather obvious, to the best of our knowledge such a crossing has not been explicitly demonstrated for coherent polaron bands before, despite the very long history of polaron studies.

Our results also demonstrate that this crossing leads to the appearance of a tilted Dirac point whose location (and very existence) can be controlled through the strength of the electron–phonon coupling. This opens the possibility of dynamically driving a system from a trivial state into one with Dirac/Weyl points that can be tuned to the Fermi energy, by resonant

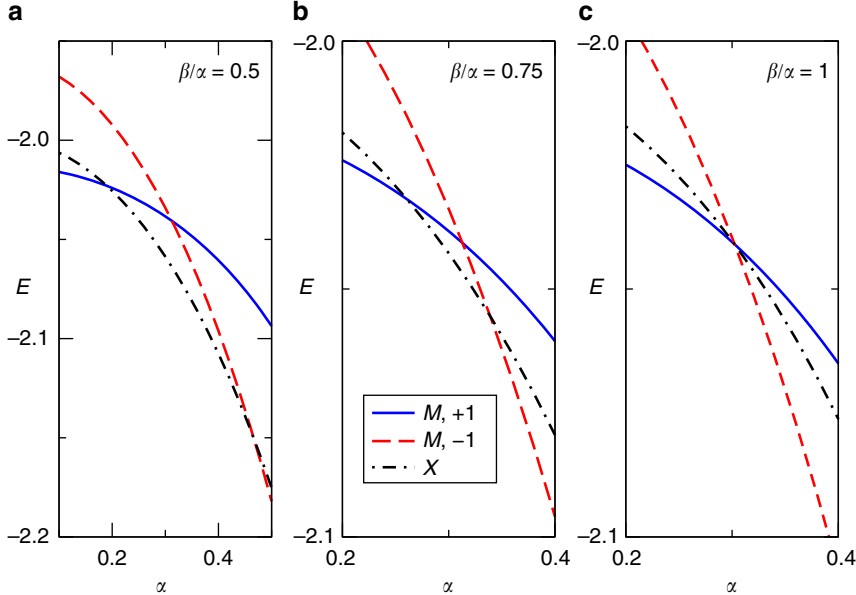

**Fig. 3** Polaron energy at high symmetry points (color online). Polaron energy at $X$ (black, dot-dash) and at $M$, for even (blue, full) and odd (red, dashed) bands, as a function of $\alpha$. **a** $\beta/\alpha = 0.5$. **b** $\beta/\alpha = 0.75$. **c** $\beta/\alpha = 1$. In all cases $t_{sp} = -1$, $t_{pp} = -0.49$, $\epsilon_s = \epsilon_p = 0$, $\Omega = 1$

pumping of appropriate phonon modes[37]. As mentioned in ref. 22, this would boost the number of phonons and effectively enhance the electron–phonon coupling to desired modes, in a way that may be more efficient than raising the temperature to generate thermal phonons. Such control may enable exploitation of topology in interesting new ways.

It is important to emphasize that this band crossing is not due to just a trivial renormalization of the bare bands because of coupling to the lattice. As discussed in ref. 36 for the 1D version of this model, this type of electron–phonon coupling not only renormalizes $t_{sp}$ and $t_{pp}$ and shifts the on-site site energies $\epsilon_s$ and $\epsilon_p$ by different amounts, but also generates longer-range $t_{ss}$, $t'_{pp}$ and $t'_{sp}$ hoppings. There is no a priori reason to expect that these changes will conspire to drive a band crossing, and in fact one can propose models where instead they push the two bands further apart. It is therefore significant that in our physical model, the electron–phonon coupling facilitates the appearance of the Dirac points. We have checked that this is the case for a wide range of parameters, even those for which the bare bands are far apart, if the electron–phonon coupling is sufficiently large.

Experimentally, there is a variety of ways to engineer Lieb lattices. They have been realized in cold atom systems[38] and photonic systems[39], and an electronic Lieb lattice has recently been realized as a molecular lattice on a Cu(111) substrate[40]. Another possible route toward the experimental realization of our model is to grow a monolayer of BaBiO$_3$ on a suitable substrate using molecular beam epitaxy. The perovskite BaBiO$_3$ is known to have fairly strong electron–phonon coupling, although likely weaker than needed for what we propose here. However, coupling to substrate phonons may significantly enhance the electron–phonon coupling within a monolayer (a similar mechanism is believed by many to be responsible for the higher $T_c$ in FeSe monolayers grown on SrTiO$_3$[41]). Moreover, doping of such thin layers is more easily achieved than is the case for 3D systems. Furthermore, it is reasonable to expect that the phenomena described here are not restricted to only the Lieb lattice in 2D, and also that similar 3D models may stabilize 3D Weyl points, but this remains to be verified. Finding suitable candidates requires detailed ab-initio calculations, beyond the scope of this work.

To conclude, we have explicitly verified that sharp crossings of polaron bands can occur in realistic models of electron–phonon coupling. They may lead to a new type of sharp GS transition, where the GS momentum remains unchanged but a symmetry of its wavefunction changes discontinuously. The crossings also result in the appearance of four type-II Dirac points along $k_x = \pm k_y$, whose position can be tuned. Our findings suggest that at least at lower energies, electron–phonon couplings can be used to control the properties of Weyl and Dirac semimetals.

## Methods

**The model.** We study a single carrier in the 2D Lieb lattice[42,43] sketched in Fig. 1b, consisting of a square lattice of ions whose valence orbital is s-type (other symmetries have qualitatively similar properties), bridged by ligands whose active orbitals are of p$_\sigma$-type. The ligand atoms are assumed to be much lighter and thus host Einstein phonons with frequency $\Omega$. Vibrations along the bond directions couple most strongly to the carrier; the other phonons will be ignored. The lattice (phonon) Hamiltonian is therefore:

$$\hat{H}_{ph} = \Omega \sum_i \left( b_{i,x}^\dagger b_{i,x} + b_{i,y}^\dagger b_{i,y} \right) \tag{1}$$

where we set $\hbar = 1$ and $b_{i,x/y}^\dagger$ creates a phonon at the corresponding p$_{x/y}$ ligand of the $i$th unit cell.

The kinetic energy of the electron is given by

$$\hat{T}_{tot} = \sum_{i,j,\gamma,\gamma'} t_{i,j}^{\gamma,\gamma'} c_{i,\gamma}^\dagger c_{j,\gamma'}. \tag{2}$$

Here $c_{i,\gamma}^\dagger$ creates an electron in the $\gamma = s$, $x$, or $y$ orbital of the unit cell centered at site $\mathbf{R}_i$. For the on-site energies $t_{i,i}^{\gamma,\gamma} = \delta_{\gamma,\gamma'}\epsilon_\gamma$ we choose $0 = \epsilon_x = \epsilon_y \equiv \epsilon_p$. We include nearest-neighbor (NN) sp hopping between a central site and its four ligands, as well as NN pp hopping between adjacent ligands. Their values $t_{i,j}^{\gamma,\gamma'} \propto \left| \mathbf{r}_{i,j}^{\gamma,\gamma'} \right|^{-n}$ depend on the interatomic distances $\mathbf{r}_{i,j}^{\gamma,\gamma'}$. For sp hopping $n = 2$, whereas for pp hopping $n = 2$ for semiconductors and $n = 3$ for transition metal oxides[44]. In a linear approximation valid for small out-of-equilibrium displacements $\hat{u}_{i,x/y} = \left( b_{i,x/y}^\dagger + b_{i,x/y} \right)/\sqrt{2M}$, we can expand $t_{i,i}^{x,y} = t_{pp}\left[ 1 - \beta\left( b_{i,x/y}^\dagger + b_{i,x/y} \right) \right]$, $t_{i,i}^{s,x} = t_{sp}\left[ 1 - \alpha\left( b_{i,x}^\dagger + b_{i,x} \right) \right]$ etc., where $t_{pp}$ and $t_{sp}$ are equilibrium hopping integrals. For an electron $t_{pp} > 0$ and $t_{sp} > 0$, while for a hole both hopping integrals become negative.

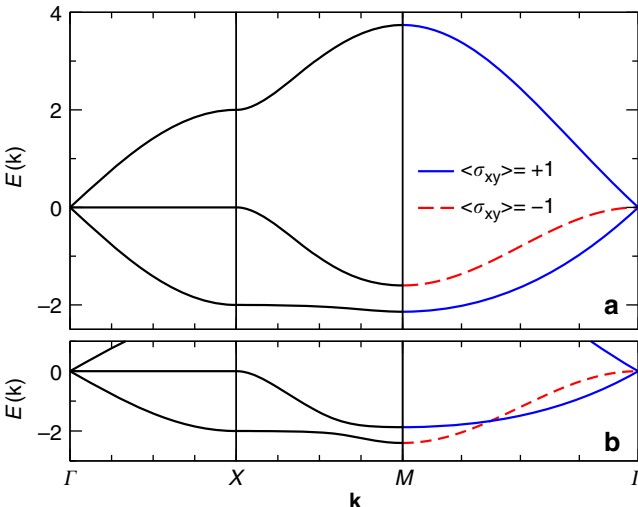

**Fig. 4** Free hole dispersion (color online). Free hole dispersion $E(\mathbf{k})$ for $t_{sp} = -1$, $\epsilon_s = \epsilon_p = 0$ and **a** $t_{pp} = -0.4$, **b** $t_{pp} = -0.6$. The $\hat{\sigma}_{xy}$ eigenvalue is indicated along the cut $\Gamma = (0, 0) \leftrightarrow M = (\pi, \pi)$. For **b** only the lower two bands which cross in a type II Dirac point are shown

Thus, $\hat{T}_{tot} = \hat{T}^{sp} + \hat{T}^{pp} + \hat{H}^{sp}_{el-ph} + \hat{H}^{pp}_{el-ph}j$. The first two terms are the hopping on the undistorted lattice:

$$\hat{T}^{sp} = -t_{sp} \sum_i c^\dagger_{i,s}\left(c_{i,x} - c_{i-\hat{x},x} + c_{i,y} - c_{i-\hat{y},y}\right) + \text{H.c.}$$

$$\hat{T}^{pp} = -t_{pp} \sum_i c^\dagger_{i,x}\left(c_{i,y} - c_{i-\hat{y},y} + c_{i+\hat{x}-\hat{y},y} + c_{i+\hat{x},y}\right) + \text{H.c.}$$

and the electron–phonon couplings are:

$$\hat{H}^{sp}_{el-ph} = \alpha t_{sp} \sum_{i,\gamma=x,y} c^\dagger_{i,\gamma}\left(c_{i,s} + c_{i+\hat{\gamma},s}\right)\left(b^\dagger_{i,\gamma} + b_{i,\gamma}\right) + \text{H.c.}$$

$$\hat{H}^{pp}_{el-ph} = \beta t_{pp} \sum_i c^\dagger_{i,x}\left(c_{i,y} - c_{i-\hat{y},y} - c_{i+\hat{x}-\hat{y},y} + c_{i+\hat{x},y}\right)$$
$$\times \left(b^\dagger_{i,x} + b_{i,x}\right) + \beta t_{pp} \sum_i c^\dagger_{i,y}\left(c_{i,x} + c_{i+\hat{y},x}\right.$$
$$\left. - c_{i+\hat{y}-\hat{x},x} - c_{i-\hat{x},x}\right)\left(b^\dagger_{i,y} + b_{i,y}\right) + \text{H.c.}$$

Choosing the lattice constant $a = 1$, we have $\alpha = 4/\sqrt{2M}$ and $\beta = n\alpha/4$.

The Hamiltonian we study is $\hat{H} = \hat{T}_{tot} + \hat{H}_{ph}$. It is important to note that it is invariant under the reflection $(x, y) \rightarrow (y, x)$, which we denote as $\hat{\sigma}_{xy}$ (inclusion of longer-range hoppings is not expected to change this, as all hoppings must obey the symmetry of the lattice). Because momenta $\mathbf{k} = (k, k)/\sqrt{2}$ are also invariant under $\hat{\sigma}_{xy}$, their corresponding eigenfunctions must be eigenfunctions of $\hat{\sigma}_{xy}$ as well (this is not true elsewhere in the BZ).

**Free carrier spectrum**. Here we briefly review the bare spectrum, in the absence of electron–phonon coupling; more details are available in ref. [45]. Hereafter the carrier is taken to be a hole. In this case, as shown in Fig. 4a, the lower two of the three bare bands are close together when $\epsilon_s \approx \epsilon_p$ and the effects of the electron–phonon coupling on the low-energy polaron bands will be more pronounced, as shown below.

The character of the bare bands is easy to infer on the $\Gamma \rightarrow M$ cut, because of invariance to $\hat{\sigma}_{xy}$. There is one odd band (red dashed line) of $p_-$ character, i.e., its eigenstate is $\propto \left(c^\dagger_{\mathbf{k},x} - c^\dagger_{\mathbf{k},y}\right)|0\rangle$, and two even bands (blue, full line) which are bonding and anti-bonding mixtures of the s and $p_+$ states. Which of these has the lowest energy at the $M$ point depends on parameters. For those of Fig. 4, namely $\epsilon_s = 0$ and $t_{pp} = -1$ (hopping integrals are negative for a hole and we set $\epsilon_p = 0$), $s + p_+$ is the lowest state. However, the $p_-$ state can be pushed below it either by increasing $|t_{pp}|$ and/or $\epsilon_s$. The two bands are degenerate at $M$ if $t_{pp} = \left(\epsilon_s - \sqrt{16t_{sp}^2 + \epsilon_s^2}\right)/8$. Further increasing $|t_{pp}|$ moves the $p_-$ band below the $s + p_+$ band in the vicinity of the $M$-point, and the point $\mathbf{k}_C = (k_C, k_C)/\sqrt{2}$ (and its symmetric counterparts) where the two bands cross is pushed toward $\Gamma$. Note that the different $\hat{\sigma}_{xy}$ eigenstates prevent band mixing so $\mathbf{k}_C$ is a sharp band-crossing point as illustrated in Fig. 4b. A straightforward analysis of the model in the vicinity of this point yields an effective Dirac Hamiltonian with a tilted cone, i.e., a type-II Dirac cone. The Dirac point is protected by the mirror symmetry of the lattice, much like Dirac points in graphene which are protected by a combination of time reversal and inversion symmetries. As in graphene or $d$-wave

superconductors we expect the Dirac points here to give rise to topologically protected edge modes that span projections of the crystal momenta $\mathbf{k}_C$ and $-\mathbf{k}_C$ onto the edge BZ. In the type-II Dirac semimetal these edge modes will overlap in energy with the bulk modes and might therefore be more difficult to observe.

**Momentum average approximation**. This is a non-perturbative, variational approximation that has been shown to provide highly accurate results at all electron–phonon coupling strengths for both Holstein[46–48] and SSH[34,35] couplings. For the 1D version of the present model, it was favorably tested against exact diagonalization in ref. [36]; the 2D version is a straightforward generalization. We implemented two different flavors of MA, with a one-site and with a two-site cloud. The latter has a bigger variational space and thus is more accurate, but differences between the two sets of results are quantitatively small, indicating that convergence has essentially been achieved. All results we show are from the two-site cloud version.

**Data availability**. All data that support the findings of this study are available from the corresponding authors upon request.

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

## Acknowledgements

We thank I. Elfimov for useful discussions. This work was supported by SBQMI, NSERC, CIfAR, and the UBC 4YF (M.M.M.).

## Author contributions

Analytical and Numerical calculations were carried out by M.M.M. with guidance from M.B.. M.M.M., M.B., G.A.S. and M.F. all contributed to the interpretation of the results and the writing of the manuscript.

## Additional information

**Competing interests:** The authors declare no competing financial interests.

