## [Peer Review File · Nature Communications]

Reviewer #1 (Remarks to the Author):

This is an interesting and important paper on the role of electron-phonon coupling in topological semimetals, a topic for which previous results are practically non-existent. It utilizes a reliable numerical method to calculate the properties of a fairly realistic model system. I recommend publication in Nature Communications after the authors consider the following suggestions:

It would help many readers if the authors defined "type-II Dirac cones" on page 1. Or at least indicated that more would be said towards the end of the paper.

In the second line of the caption for Fig. 1, (\vec{k}, ω) is repeated twice.

In Figures 1 and 2, the plotted lines are fuzzy, blurry, and almost unreadable. This suggests that the polaron quasiparticles have a finite lifetime or large imaginary part, which is not the case. They have an infinite lifetime. I am aware that the authors are trying to encode the spectral weight, but I find this of only secondary interest and a distraction. Suggestion: make the plots sharp, and move the spectral weight to a verbal description or supplementary information if the authors feel it necessary.

page 3:

"resonantly pumping phonon modes": Could the authors explain their thinking? It would be a common reaction for many readers to assume that an actively pumped system does not conserve energy in the usual sense, and does not have eigenstates or a band structure. Are the authors thinking of Floquet states? Although not emphasized by the modern Floquet industry, it would seem that in some or many situations, the Floquet eigenvalues may not be real, so there is continuous heating and not a precisely defined band structure?

spell "discontinuos"

"sharp transitions ... not possible": This perhaps overstates the case.

If you had asked whether there could be a sharp transition of the ground state from system 1 to system 2 as a function of parameters, when the two systems are completely uncoupled from each other, perhaps meters apart, the answer would of course have been yes. What the authors have pointed out is that in certain systems, there is a more elegant and natural way to accomplish this, when certain reflection symmetries exist, and the sectors of different symmetry uncouple from each other.

In the model described in Methods, the exact crucial symmetry still exists if the model is made more realistic, for example by adding further neighbor couplings?

Reviewer #2 (Remarks to the Author):

The manuscript "Type-II Dirac semimetal stabilized by electron-phonon coupling" reports a theoretical study of a 2-dimensional lattice model with the inclusion of electron-phonon interaction terms. Based on this model, it is argued that the strength of the electron-phonon interaction controls the appearance of Dirac nodes on the polaron bands. In particular, it is found that Dirac crossings of polaron bands are allowed when protected by symmetry, and it is also shown that the polaronic band minima exhibit transitions between different momenta and also between different symmetry states at the same momentum.

The methodology used appears robust, and the conclusions follow from the numerical study. Nonetheless, I have a number of points for the authors to consider:

1. It is claimed in the abstract that "We extend such studies [referring to the interplay of topology with electron-electron interaction] to interactions with the lattice". The main text begins with "One notably absent topic of discussion, in the context of topological properties of materials, is that of the interplay between topology and electron-phonon interactions." These statements are substantiated by claiming that only two papers explore these questions (Refs. 6 and 7 in the manuscript). These statements are incorrect, and reveal a poor knowledge of the relevant literature by the authors. Apart from the cited Refs. 6 and 7, there is a substantial body of work exploring the interplay between topology and electron-phonon interactions which the authors seem to be unaware of. Here I present a chronological list of selected papers in that context:

* PRB 92, 125142 (2015): it is shown how individual phonon modes can couple to electronic states to drive topological phase transitions between normal and topological insulators. The focus on individual phonon modes is along the same lines as the route taken in the present manuscript.

* PRL 115, 176405 (2015): it is shown how the electron-phonon interaction in topological insulators affects phonon linewidths and can reveal nontrivial electronic topology.

* PRL 117, 226801 (2016): it is shown how the electron-phonon interaction together with thermal expansion can drive temperature-induced topological phase transitions between normal and topological insulators.

* PRL 117, 246401 (2016): similar work to the one above on temperature-induced topological phase transitions, although ignoring the effects of thermal expansion and only including the electron-phonon interaction.

* PRB 94, 214306 (2016): it is shown how the chiral anomaly in Weyl semimetals can be detected by its effect on phonons, driven by the electron-phonon interaction.

* PRB 95, 165114 (2017): it is shown how individual phonon modes can couple to electronic states to create type-II Weyl semimetals, a similar argument to the one put forward in the present manuscript.

* Nat. Commun. 8, 14933 (2017): it is shown how the interaction between Weyl fermions and phonons drives a tunable Fano resonance in TaAs.

* arXiv:1610.03073: it is shown how the electron-phonon interaction together with the chiral anomaly in Weyl semimetals drives anomalous behaviour in phonon dispersions, optical reflectivity, and Raman scattering. This is similar to the work above in PRB 94, 214306 (2016).

I do not pretend this to be a comprehensive list, but it provides a guide of the work that has been done in this area. It appears to me that the main reason for the authors to consider Nature Communications as an appropriate journal for their work is the novelty in the study of the interplay between topology and electron-phonon interactions. In view of the literature listed above, I recommend the authors read the above papers and re-consider their work in this context.

2. In the text it is argued that the panels of Fig. 1 show the lowest energy band to move discontinuously from M to X at some α around 0.2. Unfortunately, the inset of Fig. 1 is over the X point, so this can in fact not be seen. Could the authors please re-arrange Fig. 1 to make this clear?

3. It would be interesting to have a few candidate materials whose electronic band structure and phonons might be plausibly reproduced by the 2D Lieb lattice model discussed in the present manuscript. This would make the work more interesting for experimental readers.

Reply to Reviewer 1

We are pleased that you consider our work "interesting and important" and we thank you for your useful comments. We fixed all of the spelling mistakes which you kindly pointed out and will now address the other comments in the order of their appearance.

As requested, we defined type II Dirac cones in the introduction and added some comments on what distinguishes them from regular type I Dirac cones. The relevant paragraph is the first paragraph on page 2.

As suggested, we sharpened both Figs 1 and 2. Furthermore we moved the inset in Fig. 1 in order to avoid obscuring relevant parts of the spectrum.

Regarding your comments on "resonantly pumping phonon modes." The idea is to increase the occupation number of a specific phonon mode by optically pumping the system; in turn, this increases the effect of the phonons on the electron. We suggest that this could be used to tune the position of the Dirac points, to bring them closer to the Fermi energy if they happen to be far from each other. Even though the optical pulse will push the system out of equilibrium, it is often possible to at least qualitatively understand such experiments with equilibrium techniques and the introduction of an electronic temperature. This is also how the experiment from Ref. 37 was analyzed.

Optical pumping was also mentioned in the new Ref. 18. In some ways this would be similar to an increase in temperature, which has been shown to push certain systems into a topologically non-trivial phase. However an increase in temperature is not selective when it comes to the type of phonon mode excited, and it also drives other behaviour such as thermal expansion, which may counteract the desired effect (as speculated in new Ref. [15]).

Regarding your question on the impossibility of sharp transitions. Here we are actually referring to sharp transitions in the ground state of a polaron. Until recently it was believed that such transitions are impossible. This was proved rigorously for q -dependent models of the Holstein and Froehlich-type by Lowen and Gerlach, but is not true for models where the electron-phonon coupling depends on both the quasiparticle momentum k and the transferred momentum q . We modified the text to emphasize this fact (see page 3 left column second paragraph).

The introduction of longer range hopping or other terms to make the model more realistic will not affect the reflection symmetry on the diagonal. The reason for this is that this is a symmetry of the underlying lattice and must therefore be respected by the Hamiltonian. We added a sentence to the Methods section to point this out.

We hope that we have been successful in addressing your comments and concerns and that you will continue to support the publication of our work in Nature Communications.

Reply to Reviewer 2

Dear Sir or Madam,

Thank you for your helpful comments on our manuscript and for bringing to our attention an existing body of literature on the interplay of topology and electron-phonon interactions, that we had not been aware of. The fact that a large fraction of these papers have been published within the last two years indicates that this is becoming an exciting topic in condensed matter physics. In

the following paragraphs we address your concerns and suggestions in their order of appearance.

After carefully reviewing these references and several other ones that we discovered starting from them, we concluded that our work still warrants publication in Nature Communications due to four key features which distinguish it from the existing work:

1. While other authors have focused on the effect of electron-phonon coupling on finite lifetime states located well above the bottom of the lowest band, we investigate its effect on weakly doped systems where the Fermi-level lies within the infinitely long-lived polaron bands (these are located within one optical phonon frequency of the ground-state).
2. We show how in such a case, the electron-phonon coupling can push the system into a topologically non-trivial state at zero temperature, whereas other work has focused on a temperature-assisted mechanism.
3. We demonstrate that electron-phonon interactions can shift the location of an existing Dirac point in the Brillouin Zone.
4. We point out the importance of electron-phonon interactions which depend on both the quasiparticle momentum k and the transferred momentum q . This type of electron-phonon interaction which originates from a modification of the hopping integrals is often ignored in favor of simpler interactions which only depend on q and correspond to a modification of the on-site energies. Our work shows that it is worthwhile to study the former, more complicated electron-phonon interactions because they can lead to qualitatively different physics, such as the crossing of two polaronic bands and the formation of a type II Dirac point.

Secondly, you pointed out that the inset in Fig. 1 obscured part of the spectrum. In the new figure we shifted the inset to avoid this.

Thirdly, you asked for possible candidate materials that would be modelled by a Lieb lattice like discussed in our work. This work originated from our interest in BaBiO₃, which is known to have a fairly strong electron-phonon coupling -- although likely weaker than needed for what we propose here. Of course, BaBiO₃ is also a 3D perovskite. Our model would correspond to a single layer of such a material (or any other suitable perovskite) grown on a suitable substrate, by molecular beam epitaxy. This would have the additional benefit that coupling to substrate phonons may significantly enhance the electron-phonon coupling within the layer of interest (a similar mechanism is believed by many to be responsible for the higher T_c in FeSe monolayers grown on SrTiO₃, for instance). Moreover, doping of such thin layers is also more easily achieved, than is the case for 3D systems.

Based on such elementary considerations, we believe that finding good candidates to verify the physics we discuss, is feasible. This being said, we prefer not to speculate on them in the current work, because a reasonable proposal should involve DFT-type calculations to see which layers may actually be stable on which substrates. Doing such an analysis is well beyond the scope of the current work. Moreover, note that here we showed that Lieb lattices exhibit this physics, but further study may reveal even more suitable lattice structures, both in 2D and possibly also in 3D systems.

Reviewer #1 (Remarks to the Author):

The authors have adequately addressed most of the issues that were raised in my first referee report, and I recommend publication of the revised manuscript, with two caveats:

(1) On my printed copy, Fig. 2 still contains only the faintest ghosts of lines. I am not sure whether they are there or not. The figure on my computer screen is only slightly better. It may be the most realistic for a line with 20 times less spectral weight to be 20 times fainter, except that the reader cannot see it. If the authors are determined to display the spectral weight like this, I would suggest something like mapping the spectral weight range $[0,1]$ into a line thickness range of something like $[0.3, 1]$ so that nothing disappears from view.

(2) The second referee included a number of references of which the authors were unaware, and I was also unaware. I have not read all of these new references, and I leave it to the second referee to determine whether they are addressed adequately in the revised version, and whether there is sufficiently small overlap with the present work.

Reviewer #2 (Remarks to the Author):

The authors have addressed the comments posed by referees 1 and 2, but not those of referee 3. Furthermore, from their responses to my original points:

1. The authors have changed their argument as to why the paper merits publication in Nature Communications from a study initiating the field of electron-phonon interaction in topological materials in the original manuscript to "Here, we open a new area of research by showing that for weakly doped systems, sufficiently strong electron-phonon coupling due to the modulation of hopping integrals, may drive a system into a topologically non-trivial state even at $T = 0$." in the revised manuscript in response to the new body of literature that I pointed out to them in my original review. I imagine that the concept of "new area of research" is rather subjective, and I am still unsure that the present manuscript is appropriate for Nature Communications. However, if I were in the minority on this point, I would not oppose publication.

2. The authors described potential material realizations of the phenomenology they propose in the referee response letter, but argued that they did not want to include it in the main manuscript because they did not have supporting DFT calculations. I still think that something along the lines of what appears in their referee response letter would be interesting to the community, in particular to DFT practitioners who might pick up on this and carry out materials simulations, and to the experimental community. This view appears to be supported by that of referee 3, whose comments were not addressed by the authors.

Reviewer #3's comments in the last round review:

In this paper the authors study the topological properties of polarons, i.e., electronic quasiparticles dressed by phonon clouds. By considering electrons hopping on a Lieb lattice coupled to Einstein phonons, they show that polaron bands can exhibit a protected crossing, forming a type-II Dirac cone. They argue that these Dirac polarons lead to strong signatures in quantum oscillation measurements.

The paper is reasonably well written and not too difficult to follow. However, in my opinion, it does not rise to the level required by Nature Communications. It is unclear how generic these Dirac polarons are. The authors have shown its existence in only one particular model system and have not even worked out the topological invariant that protects it. What are the general requirements for this Dirac polaron to occur? Moreover, in order to be taken seriously as an experimentally relevant proposal the authors should (1) suggest material systems where this can occur, (2) compute surface states and topological invariants, and (3) work out other experimental signatures.

Based on the above points, I cannot recommend the paper for publication in Nature Communications.

Reply to Reviewer 1

Thank you for your helpful comments. We are happy to hear that you recommend publication of our manuscript in Nature Communications. In response to your request we have made the lines in Fig. 2 less faint by changing the scaling of the color map and in some cases choosing a different orbital projection for the spectral function.

Reply to Reviewer 2

Thank you for your helpful comments. As you point out the interplay of topology and electron-phonon coupling has seen a recent increase in attention. Nevertheless, for the reasons stated in our previous reply and the introduction of our manuscript, we believe that the focus on polaronic bands and medium to strong electron-phonon coupling gives our work the novelty required for publication in Nature Communications. We also note that Reviewer 1 supports publication of our manuscript in Nature Communications.

Regarding experimental realizations of our model, as you suggested we included a paragraph on this in the revised version of our manuscript. The paragraph can be found at the very end of the Results section, just before the concluding paragraph.

With regards to Reviewer 3's comments, we were only made aware of these after our previous reply and therefore did not have a chance to address them. Reviewer 3 asks that we calculate the topological constant which protects the Dirac cones. However, the Dirac points in our model are protected by symmetry, not by a topological invariant. The situation is analogous to graphene, where point nodes are protected by a combination of time reversal and inversion symmetry. In our case point nodes are protected by mirror symmetry along the diagonal. As long as that symmetry is present no perturbation can remove these nodes (but they can pairwise annihilate at high symmetry points in the BZ). We added some clarifying comments to the Methods section (page 4 left column, last paragraph).

With regard to edge states we expect the Dirac points to give rise to topologically protected edge modes, similar to graphene (these are the well-known flat bands associated with the zig-zag edges in graphene). In a type-II Dirac situation these edge modes might however be difficult to observe because unlike in type-I Dirac systems, they would typically overlap in energy with the bulk states. Again we added some clarifying comments to the Methods section (page 4 left column, last paragraph).

We hope that given these changes you will support publication of our manuscript in Nature Communications.

Reply to Reviewer 3

Thank you for your helpful comments on our manuscript. You oppose publication of our manuscript for several reasons, to all of which we answer in the following.

The first point is that it is not clear how generic "these Dirac points are". As we have shown in our work the interplay between topology and strong-electron phonon coupling is quite intricate. Therefore it is difficult to make general statements, as one needs to rely on numerics. However, we have shown that in a model with the required symmetry properties for the formation of type-II Dirac points, electron-phonon coupling can help stabilize these Dirac points. Even though this is non-trivial it suggests that the same should be possible in other models. As we point out at the end of the Results section of the revised manuscript it is quite likely that the physics discussed in our work is not restricted to Lieb lattices and/or 2D.

Regarding the topological invariant, this is actually a misunderstanding. In our model the Dirac points are protected by symmetry, not a topological invariant. The situation is analogous to graphene, where point nodes are protected by a combination of time reversal and inversion symmetry. In our case point nodes are protected by mirror symmetry along the diagonal. As long as that symmetry is present no perturbation can remove these nodes (except that they can pairwise annihilate at high symmetry points in the BZ). We added some clarifying comments to the Methods section (page 4 left column, last paragraph).

With regard to edge states we expect the Dirac points to give rise to topologically protected edge modes, similar to graphene (these are the well-known flat bands associated with the zig-zag edges in graphene). In a type-II Dirac situation these edge modes might however be difficult to observe because unlike in type-I Dirac systems they would typically overlap in energy with the bulk states. Again we added some clarifying comments to the Methods section (page 4 left column, last paragraph).

You furthermore asked that we suggest experimental realizations for our model, which are included in the revised manuscript. The corresponding paragraph can be found at the end of the Results section, just before the concluding paragraph.

Finally let us comment on the experimental signatures of type-II Dirac/Weyl points. We have added some clarifying comments to the first paragraph of the introduction which explain the difference between type-II Dirac/Weyl cones and ordinary type-I Dirac/Weyl cones. From this it should be clear that the physics of type-II Dirac/Weyl cones and consequently their experimental signatures are different from those of type-I Dirac/Weyl points. Since our work is mainly concerned with the stabilization of type-II Dirac points due to electron-phonon coupling we do not wish to go into too much detail on the experimental signatures. However, the interested reader will find more information in the papers which we cite, specifically Refs 5-8.

We believe that the revised manuscript is significantly improved over the previous version and hope that together with the clarifications provided above you will find it suitable for publication in Nature Communications.